# Anthropometric and biochemical nutritional indicators and survival in women with breast cancer: A retrospective cohort study

Lourdes Sánchez-Saldaña[1], Michelle Lozada-Urbano[2], Yasser Sullcahuaman-Allende[1], José Cotrina-Concha[1], Marco Velarde-Méndez[1], Jorge Chavez-Chocano[1], Enrique Rodríguez-Coyla[1], Luis Zambrano-Jaimes[1], Raúl Mantilla-Quispe[1], Jaime Rosales-Rimache [ID][3]*

**1** Instituto Nacional de Enfermedades Neoplásicas, Lima, Perú, **2** Universidad Privada Norbert Wiener, Lima, Perú, **3** Universidad Científica del Sur, Carrera de Medicina, Lima, Perú

* jrosalesr@cientifica.edu.pe

## Abstract

### Introduction

Weight gain has been observed in breast cancer (BC) survivors, and this can affect survival and lead to adverse health effects such as overweight and obesity. It may be associated with an increase in cancer recurrence of between 35 and 40% with worse survival results, especially in those with estrogen receptor-positive breast cancer. Therefore, this study aims to determine the association between nutritional status and survival in women with BC at a Peruvian Specialized Oncology Institute.

### Methods

This retrospective cohort involves a sample of 195 breast cancer patients whose medical records were obtained from 2017. Survival differences were measured using Cox proportional hazards models, expressed as hazard ratios (RR) and 95% confidence intervals. Univariate and multivariate analyses were performed previously.

### Results

We show no association between anthropometric variables; however, a relation was found with biochemical variables, including iron [HR: 2.61, CI 95%: 1.23–5.55, p = 0.013], albumin [HR: 10.02, CI 95%: 2.86–35, p = 0.0001], and total lymphocyte count [HR: 2.12, CI 95%: 1.00–4.50, p = 0.045] with overall survival.

### Conclusion

We conclude that while no association was found between anthropometric variables and survival in women with breast cancer, specific biochemical markers. Further

**Data availability statement:** There is no supplementary data associated with this study.

**Funding:** The author(s) received no specific funding for this work.

**Competing interests:** The authors have declared that no competing interests exist.

research is needed to adapt the recommendations for food quality control included in the diet and assess whether it leads to better outcomes.

## Introduction

BC is a neoplastic disease originating in breast tissue, typically arising in the ducts or lobules. Consequently, these types are called ductal and lobular carcinoma [1]. It has the highest incidence rate worldwide, with 47.8 cases per 100,000 people and 2.2 million new cases diagnosed annually [2]. The five-year survival rate has increased from 63% to 90% from the 1960s to recent years respectively [3]. However, those who have survived continue to be at high risk of recurrence, even up to 20 years after diagnosis [4], and with a high probability of weight gain and comorbidities such as cardiovascular and metabolic changes [5].

Risk factors for BC include age, hereditary predisposition, and environmental factors such as poor dietary habits (high intake of unhealthy foods and low intake of fruits and vegetables), high intake of alcoholic beverages, smoking, low physical activity levels, and nulliparity [6]. A healthy lifestyle, including weight control and a quality diet, reverses the risk and effect of breast cancer. Foods like processed meats and saturated fats increase risk factors, while fiber, ω-3 PUFAs, and antioxidants may protect by reducing inflammation and oxidative stress [7]. Evidence establishes that the relationship between obesity and breast cancer prognoses is associated with the cancer subtype [8]. BC risk is associated with overweight and obesity, specifically with hormone receptor-positive type [9–11].

Author Wei and colleagues propose a prognostic model for BC that incorporates nutritional and inflammatory biomarkers such as lymphocytes, platelet count, haemoglobin levels, albumin-globulin ratio, prealbumin level, together with clinical-pathological characteristics to predict survival and disease-free status in breast cancer [12]. Similarly, other studies have also used inflammatory and nutritional markers, such as neutrophil, lymphocyte, platelet, serum ferritin, and lymphocyte/monocyte ratios, as well as the prognostic nutritional index of haemoglobin, albumin, transferrin, albumin/globulin, and prealbumin, which have been shown to be clinical prognostic predictors for various types of cancer [13,14].

Obesity in premenopausal women is linked to a higher risk of aggressive triple-negative breast cancer (TNBC). While higher BMI consistently correlates with worse outcomes in estrogen receptor-positive breast cancer, results are mixed for other subtypes. Recent findings show that BMI's impact on outcomes is independent of subtype classification using the PAM50 assay [8]. A meta-analysis that included 213,075 BC describes that for each increase of 5 kg/m² in BMI, the risk of general mortality and specific mortality increased by 17% and 29%, respectively [15]. In addition to BMI, waist-hip ratio has also shown a significant positive association with BC mortality in postmenopausal women [16].

Our study aims to determine the association between anthropometric variables (BMI, Waist Circumference, Triceps Skinfold Thickness, Moderate caloric malnutrition, Arm Muscle Circumference, Glucose, Hemoglobin, Albumin, and Total

Lymphocyte count), and survival expressed as the follow-up period from the date of the operation to the date of death or the date of the last follow-up in women with breast cancer at a Peruvian Specialized Oncology Institute.

In that sense, we ask ourselves: There is a significant association between anthropometric variables (Body Mass Index, Waist Circumference, Triceps Skinfold Thickness, Moderate Caloric Malnutrition, Arm Muscle Circumference) and biochemical indicators (Glucose, Hemoglobin, Albumin, and Total Lymphocyte Count) with overall survival, expressed as the follow-up period from the date of surgery to the date of death or last follow-up, in women with BC treated at a specialized oncology institute in Peru.

## Methodology

### Study design and population

We designed a single-arm retrospective cohort based on adult women with breast cancer (BC) in the pre-surgical stage and treated at the National Institute of Neoplastic Diseases (INEN) in Peru in April 2017. This institution is Peru's leading specialized center for evaluating individuals with oncological diseases. The diagnosis of BC was conducted according to the INEN Clinical Practice Guideline, approved by RJ No. 650–2013-J-INEN [17]. We included195 records of women with BC between 18 and 60 years who were followed for 64 months from the time of initial diagnosis until their death. They were monitored through anthropometric, biochemical, and nutritional indicators tests. Women with breast diseases other than breast cancer were excluded. The clinical history review was conducted in the INEN clinical history archive between January 15 and March 29, 2024.

### Variables

Outcome: Overall survival rate.

Independent variables: Body Mass Index (BMI), waist circumference, triceps skinfold thickness, arm muscle circumference, and biochemical results, including glucose, hemoglobin, albumin, and total lymphocyte count. The anthropometric parameters were performed and interpreted by experienced individuals.

We considered information about BC subtypes based on molecular and immunohistochemical analysis. This method allows for a more accurate classification and correlates with these patients' quality of life, survival, and relapse rates. We used biomarkers associated with estrogen, progesterone, and HER2 receptors. Additionally, the histological grade of tumor proliferation, indicated by Ki-67, was considered. Ki-67 is a protein expressed during cell division; a high percentage of this marker indicates a high tumor growth rate [18].

| Molecular subtype | Estrogenic receptors (ER) and progesterone receptors (PR) | HER2 | Proliferation index (Ki-67) |
|---|---|---|---|
| Luminal A | ER positive, PR positive | HER2 negative | proliferative Ki-67 low |
| Luminal B | ER positive, PR negative/low | HER2 negative | proliferative Ki-67 high |
| Her2 | ER negative, PR negative | HER2 positive | proliferative Ki-67 varied |
| Triple negative | ER negative, PR negative | HER2 negative | proliferative Ki-67 high |

Index: Low Ki-67 < 14% / High Ki-67 ≥ 14%.

### Statistical analysis

A descriptive analysis of qualitative variables was made using frequencies and percentages, and a summary analysis of quantitative variables was performed using measures such as average, standard deviation, minimum, maximum, median, quartile 1 and quartile 3.

Anthropometric and biochemical variables were grouped into intervals and assessed as follows: BMI (16 to <17: Underweight Grade II, 17 to <18.5: Underweight Grade I,18.5 to <25: Normal, 25 to <30: Overweight, 30 to <35: Obesity Grade I, 35 to <40: Obesity Grade II, ≥ 40: Obesity Grade III); Waist Circumference (<80 cm Low risk, ≥ 80 cm High risk, ≥ 88 cm Very

high risk); Triceps Skinfold Thickness (<60% Severe caloric malnutrition, 60–79% Moderate caloric malnutrition, 80–89% Mild caloric malnutrition); Arm Muscle Circumference (<60% Severe protein malnutrition, 60–79% Moderate protein malnutrition, 80–89% Mild protein malnutrition); Glucose (<70 Low, 70–110 Optimal, >110 High); Hemoglobin (<8 Severe anemia, 8–10.9 Moderate anemia, 11–11.9 Mild anemia, ≥12 No anemia); Albumin (<21 Severe visceral malnutrition, 21–27 Moderate visceral malnutrition, 28–34 Mild visceral malnutrition, >34 Normal) and Total Lymphocyte Count (<800 Severe immunosuppression, 800–1199 Moderate immunosuppression, 1200–1500 Mild immunosuppression, >1500 Normal).

In order to estimate overall survival, the follow-up period ranged from the date of surgery to the date of documentation of death (event of interest) or the date of the last follow-up. Patients who did not reach the event of interest were considered censored. Overall survival estimates were obtained using the Kaplan-Meier method, and differences in survival based on the variables under study were assessed with the log-rank test. Categories of variables showing an imbalance in the number of cases per category were grouped. An univariates and multivariate Cox proportional hazards models were fitted with the variables that showed significant differences in overall survival to evaluate their effect on the risk of death; the proportional hazards assumption was tested in the adjusted model.

A p-value <0.05 was considered for a significant difference in overall survival and a significant effect on the risk of dying. The analyses were conducted using R software.

### Ethical aspects

The study was approved by the Institutional Research Ethics Committee of the National Institute of Neoplastic Diseases (INEN) in Peru with Letter N° 074–2023-CRPI-DI-DICON/INEN, dated December 15, 2023. The information obtained from the study was stored in a database with restricted access for the INEN nutrition research team. All samples were coded to maintain patient anonymity. No procedure in this study interfered with or compromised the therapeutic management of patients.

## Results

We evaluated 195 patients and there were no records excluded in this study.

### Demographic and clinical variables

Table 1 shows the demographic and clinical variables. The average age of the patients was 47 years (range: 20–78 years). The marital status variable indicates that the highest percentage were single patients, 78 (40.0%), followed by married patients, 71 (36.4%). Regarding education level, 75 (38.5%) patients had incomplete secondary education, and 33 (16.9%) had university studies. Most patients were from the Lima region, 86 (44.1%).

In terms of histological type, 136 (69.7%) patients had infiltrating ductal carcinoma, 7 (3.6%) had other histological types, and 52 (26.7%) patients had no record of histological type. According to estrogen receptor status, 56 (28.7%) patients had negative estrogen receptors, and 108 (55.4%) had positive estrogen receptors. For progesterone receptor status, 63 (32.3%) patients had negative progesterone receptors, and 102 (52.3%) had positive progesterone receptors.

Regarding HER2 status, 92 (47.2%) patients had HER2-negative status, and 75 (38.5%) had HER2-positive status. For Ki-67, 21 (10.8%) patients had a Ki-67 of 14% or less, and 147 (75.4%) had a Ki-67 greater than 14%.

About type of surgery, 40 (20.5%) patients underwent breast-conserving surgery, and 155 (79.5%) underwent mastectomy. For the stage, there were 8 (4.1%) patients in Stage I, 62 (31.8%) in Stage II, 70 (35.9%) in Stage III and 9 (4.6%) in Stage IV.

### Anthropometric variables

Table 2 shows summary measurements of the anthropometric variables of 195 patients under the study. The average weight was 64.3 kg (range: 30.5 to 102 kg). The average height was 1.53 m (range: 1.36 to 1.71 m). The average BMI was

**Table 1. Distribution of patients according to demographic, clinical and surgery treatment variables.**

|  | N = 195 |
|---|---|
| **Age, years** | |
| Average (SD) | 47 (9.49) |
| **Marital status** | |
| Single | 78 (40.0%) |
| Partner | 39 (20.0%) |
| Married | 71 (36.4%) |
| Divorced | 4 (2.1%) |
| Widow | 3 (1.5%) |
| **Level of education** | |
| No education | 2 (1.0%) |
| Incomplete elementary school | 30 (15.4%) |
| High school | 99 (50.8%) |
| Technical studies | 29 (14.9%) |
| University studies | 33 (16.9%) |
| No Registration | 2 (1.0%) |
| **Place of birth** | |
| Lima | 86 (44.1%) |
| Regions outside Lima | 109 (55.9%) |
| **Histological type** | |
| Infiltrating ductal carcinoma | 136 (69.7%) |
| Others | 7 (3.6%) |
| No registration | 52 (26.7%) |
| **Estrogen receptor status** | |
| Negative | 56 (28.7%) |
| Positive | 108 (55.4%) |
| No registration | 31 (15.9%) |
| **Progesterone receptor status** | |
| Negative | 63 (32.3%) |
| Positive | 102 (52.3%) |
| No registration | 30 (15.4%) |
| **HER2 status** | |
| Negative | 92 (47.2%) |
| Positive | 75 (38.5%) |
| No registration | 28 (14.4%) |
| **Ki-67** | |
| ≤14% | 21 (10.8%) |
| >14% | 147 (75.4%) |
| No registration | 27 (13.8%) |
| **Type of surgery** | |
| Breast-conserving | 40 (20.5%) |
| Mastectomy | 155 (79.5%) |
| **Stage** | |
| I | 8 (4.1%) |
| II | 62 (31.8%) |
| III | 70 (35.9%) |
| IV | 9 (4.6%) |

*(Continued)*

**Table 1.** (Continued)

|  | N = 195 |
|---|---|
| No registration | 46 (23.6%) |

SD: standard deviation.

HER2 = Estrogen receptor (-) and progesterone receptor (-) Her 2 (+++)

Ki 67 (varied).

**Table 2. Summary measures of anthropometric variables (N = 195).**

|  | Average | SD | Min | Max | Median | Q1 | Q3 |
|---|---|---|---|---|---|---|---|
| **Weight, kg** | 64.3 | 12.49 | 30.5 | 102 | 63.5 | 56 | 70.35 |
| **Height, m** | 1.53 | 0.07 | 1.36 | 1.71 | 1.53 | 1.483 | 1.57 |
| **BMI, kg/m²** | 27.5 | 5.28 | 16.49 | 47.2 | 26.71 | 24.29 | 30.39 |
| **Waist circumference, cm** | 92.7 | 10.45 | 63 | 133 | 92.5 | 86.75 | 99 |
| **Triceps skinfold thickness, cm** | 24.42 | 6.77 | 5 | 46 | 24.5 | 20 | 28 |
| **Mean arm circumference, cm** | 30.89 | 4.59 | 19.5 | 49 | 30.5 | 28 | 33 |
| **Arm muscle circumference, cm** | 23.2 | 3.55 | 14.389 | 38.261 | 22.704 | 21.153 | 24.789 |

SD: standard deviation, Min: minimum, Max: maximum, Q1: quartile 1, Q3: quartile 3.

27.5 kg/m2 (range, 16.49 to 47.2 kg/m2). The average waist circumference was 92.7 cm (range: 63–133 cm). The average triceps skinfold thickness was 24.42 cm (range: 5–46 cm). The average value of the mean arm circumference was 30.89 cm (range: 19.5 to 49 cm). The average arm muscle circumference was 23.2 cm (range, 14.39 to 38.26 cm).

According to BMI levels, 78 patients (40%) had a normal BMI, 78 patients (40.0%) were overweight, 36 patients (18.5%) had grade I obesity, 14 patients (7.2%) had grade II obesity, and four patients (2.1%) had grade III obesity (Fig 1). Based on waist circumference assessment, 138 patients (70.8%) with very high risk had a higher waist circumference (Fig 1). According to the triceps skinfold thickness assessment, 181 patients (92.8%) were not found to have caloric malnutrition (Fig 1). Based on arm muscle circumference assessment, 154 patients (79.0%) were not found to have protein malnutrition (Fig 1).

## Biochemical variables

Table 3 shows summary measurements of the biochemical variables of 195 patients in the study. The average glucose level was 94.1 g/dL (range: 66.6 to 340.2 g/dL). The average hemoglobin was 12.5 g/dL (range: 7.3 to 14.8 g/dL). The average albumin was 47.6 g/L (range, 4.3 to 462 g/L). The average total lymphocyte count was 1945.4 (range: 410–5890). Glucose levels showed 179 patients (91.8%) with an optimal level (Fig 2). Hemoglobin assessment revealed 19 patients (9.7%) with moderate anemia, 41 patients (21.0%) with mild anemia, and 134 patients (68.7%) without anemia (Fig 2). Albumin assessment showed 154 patients (79.0%) with a normal level (Fig 2). Total lymphocyte count assessment indicated 143 patients (73.3%) with an expected count and 26 patients (13.3%) with mild immunosuppression (Fig 2).

## Overall survival

34 (17.4%) deaths were documented among the total number of patients under study, with a median follow-up time for estimating overall survival of 64 months and a range from 1 month to 75 months. Overall survival at 12, 36, and 60 months was estimated to be 93%, 84%, and 79%, respectively (Table 4). Significant differences in overall survival were found according to hemoglobin, albumin and total lymphocyte count levels.

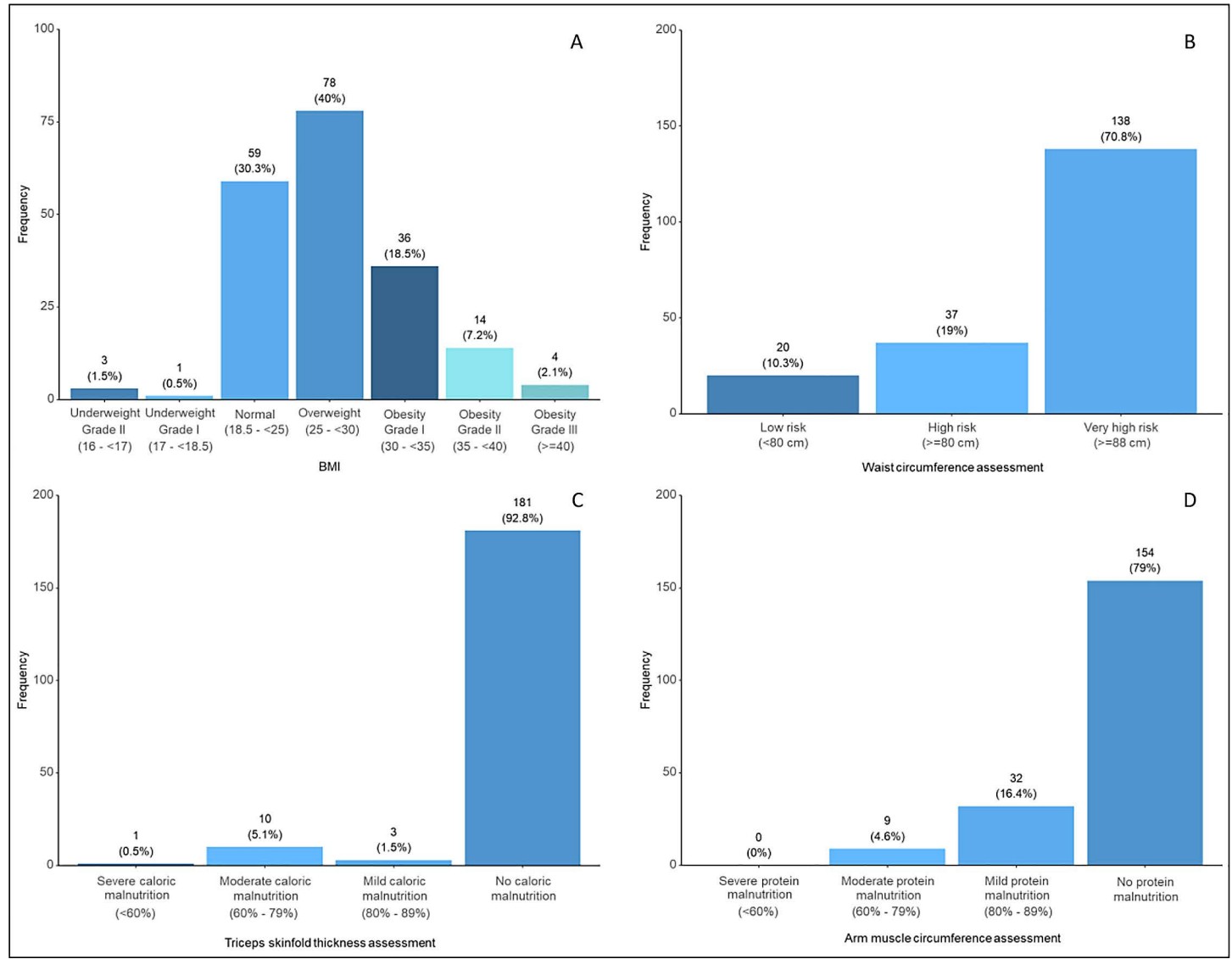

**Fig 1. Distribution of patients according to anthropometric indicators.** Distribution of the study patients (n = 195) according to body mass index (BMI) levels **(A)**, waist circumference classifications **(B)**, triceps skinfold thickness classifications **(C)**, and arm muscle circumference classifications **(D)**.

**Table 3. Summary measures of biochemical variables.**

|  | N | Average | SD | Min | Max | Median | Q1 | Q3 |
|---|---|---|---|---|---|---|---|---|
| **Glucose, mg/dL** | 193 | 94.1 | 23.64 | 66.6 | 340.2 | 90 | 84.6 | 99 |
| **Hemoglobin, g/dL** | 195 | 12.5 | 1.26 | 7.3 | 14.8 | 12.6 | 11.7 | 13.4 |
| **Albumin, g/L** | 158 | 47.6 | 45.51 | 4.3 | 462 | 43.45 | 40.5 | 45.5 |
| **Total lymphocyte count, cel/µL** | 195 | 1945.4 | 689.02 | 410 | 5890 | 1900 | 1480 | 2350 |

SD: standard deviation, Min: minimum, Max: maximum, Q1: quartile 1, Q3: quartile 3.

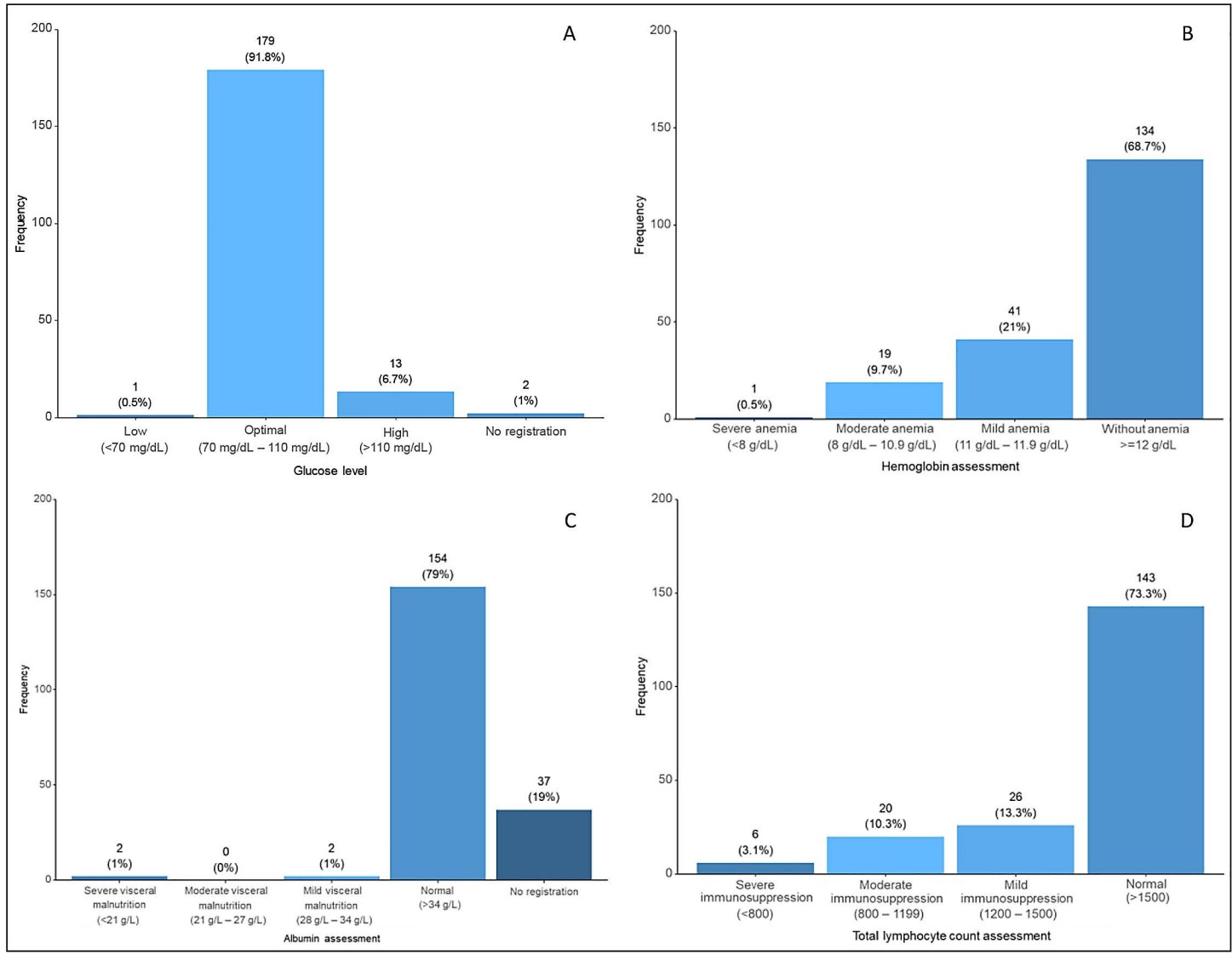

**Fig 2. Distribution of patients according to biochemical and hematological indicators.** Distribution of the study patients (n = 195) according to glucose levels **(A)**, hemoglobin classifications **(B)**, serum albumin classifications **(C)**, and total lymphocyte count classifications **(D)**.

Fig 3 shows that the lifespan of patients without anemia is longer than that of patients with anemia, but the median overall survival time could not be estimated for either group. Similarly, the figure indicates that patients with normal albumin level longer than those with visceral malnutrition. The median overall survival for the visceral malnutrition was 10.5 months, whereas it could not be estimated for the group with a normal albumin level. Additionally, Fig 3 shows that patients with a normal total lymphocyte count have a longer lifespan than those with immunosuppression, but the median overall survival time could not be estimated for either group.

Table 5 shows the estimated effect on the risk of death for the characteristics that demonstrated significant differences in over-all survival. When evaluated together, hemoglobin, albumin, and total lymphocyte count levels significant affect the risk of death. There is an increased risk of death in patients with anemia, with an adjusted hazard ratio (HR) of 2.61 (95% CI: 1.23–5.55). There

**Table 4. Estimate of overall survival (OS) according to study variables.**

| | N (events) | OS | | | p-value [a] |
|---|---|---|---|---|---|
| | | 12m | 36m | 60m | |
| **All patients** | 195 (34) | 93% | 84% | 79% | |
| **Age groups** | | | | | |
| ≤40 years | 48 (6) | 98% | 85% | 85% | |
| >40 years | 147 (28) | 92% | 84% | 77% | 0.307 |
| **Type of surgery** | | | | | |
| Breast-conserving | 40 (4) | 97% | 90% | 86% | |
| Mastectomy | 155 (30) | 92% | 82% | 77% | 0.287 |
| **Stage** | | | | | |
| I | 8 (2) | 100% | 71% | 71% | |
| II | 62 (7) | 96% | 88% | 85% | |
| III | 70 (19) | 87% | 77% | 71% | |
| IV | 9 (3) | 100% | 75% | 56% | 0.206 |
| **Body Mass Index (BMI)** | | | | | |
| Underweight (<18.5) | 4 (1) | 67% | 67% | 67% | |
| Normal (18.5 – <25) | 59 (8) | 92% | 83% | 83% | |
| Overweight (25 – <30) | 78 (18) | 92% | 80% | 73% | |
| Obesity (>=30) | 54 (7) | 98% | 92% | 84% | 0.385 |
| **Waist circumference levels** | | | | | |
| Low risk (<80 cm) | 20 (5) | 77% | 71% | 71% | |
| High risk (>=80 cm) | 37 (7) | 94% | 81% | 77% | |
| Very high risk (>=88 cm) | 138 (22) | 95% | 87% | 81% | 0.397 |
| **Triceps skinfold thickness levels** | | | | | |
| Caloric malnutrition (<=89%) | 14 (4) | 69% | 69% | 69% | |
| No caloric malnutrition | 181 (30) | 95% | 85% | 80% | 0.141 |
| **Arm muscle circumference levels** | | | | | |
| Muscle protein malnutrition (<=89%) | 41 (7) | 92% | 79% | 79% | |
| No protein malnutrition | 154 (27) | 94% | 85% | 79% | 0.885 |
| **Glucose levels** | | | | | |
| Low (<70 mg/dL) | 1 (0) | 100% | 100% | 100% | |
| Normal (70 mg/dL – 110 mg/dL) | 179 (33) | 93% | 83% | 78% | |
| High (>110 mg/dL) | 13 (1) | 92% | 92% | 92% | 0.615 |
| **Hemoglobin levels** | | | | | |
| Anemia (<12 g/dL) | 61 (17) | 88% | 73% | 67% | |
| Without anemia (>=12 g/dL) | 134 (17) | 96% | 89% | 85% | **0.0058** |
| **Albumin levels** | | | | | |
| Visceral malnutrition (<=34 g/L) | 4 (3) | 25% | 25% | 25% | |
| Normal (>34 g/L) | 154 (27) | 94% | 84% | 79% | **0.0001** |
| **Total lymphocyte count levels** | | | | | |
| Immunosuppression (<=1500 cel/µL) | 52 (14) | 88% | 72% | 70% | |
| Normal (>1500 cel/µL) | 143 (20) | 95% | 88% | 83% | **0.029** |

[a]Log-rank test.

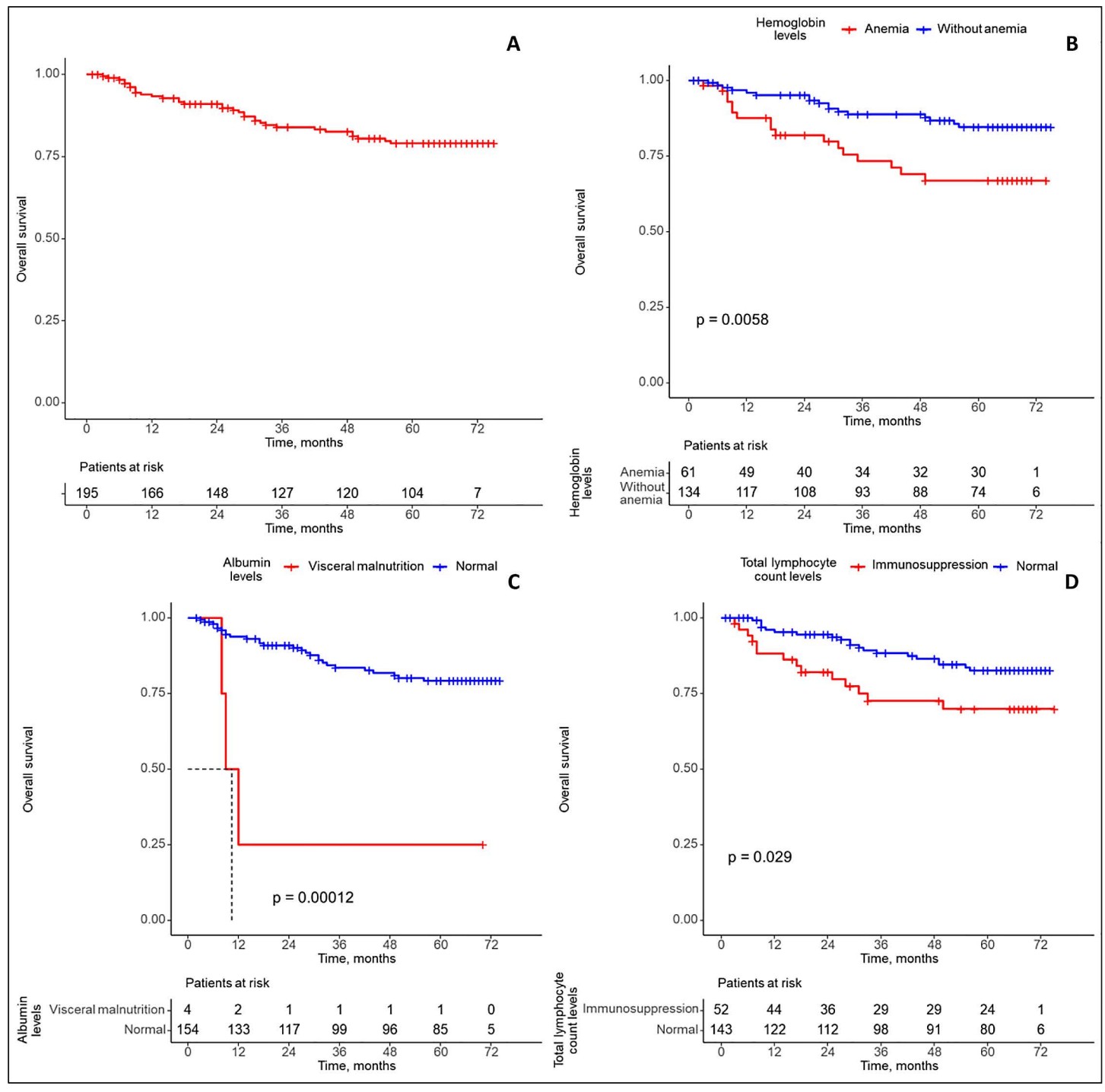

**Fig 3. Overall survival and survival according to nutritional indicators.** Overall survival of the patients included in the study (A) and survival stratified according to hemoglobin levels **(B)**, serum albumin levels **(C)**, and total lymphocyte count levels **(D)**, assessed using the Kaplan–Meier method.

**Table 5. Biochemical indicators associated with BC survival in a Cox regression analysis.**

| | Univariate | | | Multivariate | | |
|---|---|---|---|---|---|---|
| | HR | 95% CI | p-value | HR | 95% CI | p-value |
| **Hemoglobin levels** | | | | | | |
| Without anemia | Ref. | | | Ref. | | |
| Anemia | 2.50 | 1.27, 4.89 | **0.008** | 2.61 | 1.23, 5.55 | **0.013** |
| **Albumin levels** | | | | | | |
| Normal | Ref. | | | Ref. | | |
| Visceral malnutrition | 7.62 | 2.28, 25.46 | **0.001** | 10.02 | 2.86, 35.0 | **<0.001** |
| **Total lymphocyte count levels** | | | | | | |
| Normal | Ref. | | | Ref. | | |
| Immunosuppression | 2.11 | 1.06, 4.17 | **0.033** | 2.12 | 1.00, 4.50 | **0.049** |

HR: Hazard Ratio, CI: confidence interval, Ref.: reference.

is an increased risk of death for patients with visceral malnutrition, with an adjusted HR of 10.0 (95% CI: 2.86–35.0). There is an increased risk of death in patients with immunosuppression, with an adjusted HR of 2.12 (95% CI: 1.00–4.50).

## Discussion

Breast cancer (BC) survival is influenced by multiple clinical, biological, and lifestyle actors, among which nutritional markers emerged as potential prognostic determinants. In the present study, although anthropometric indicators such as BMI, waist circumference, triceps skinfold thickness, and arm muscle circumference did not show a statistically significant association with overall survival, biochemical markers, specifically hemoglobin and albumin, demonstrated a significant relationship with survival outcomes.

Previous evidence supports the relevance of nutritional status in BC prognosis. Vance et al. [19], reported that weight gain is frequent among women receiving adjuvant chemotherapy, particularly those with prolonged treatment duration and premenopausal status, leading to unfavorable changes in body composition characterized by decreased lean mass and increased adiposity. Similarly, large cohort studies have shown that elevated BMI is associated with reduced breast cancer–specific survival, particularly among estrogen receptor–positive patients (HR = 1.11). Morra et al. [20], in a pooled analysis of 67 studies including over 120,000 women, found that obesity (BMI ≥ 30 kg/m²) was significantly associated with increased all-cause mortality (HR = 1.19; 95% CI = 1.06–1.34). These findings reinforce the hypothesis that adiposity, through endocrine and inflammatory pathways, may influence tumor progression and treatment response.

Conversely, some studies have failed to demonstrate such associations. For example, analyses of body composition via computed tomography in over 3,000 women with BC revealed no relationship between BMI or body composition parameters and all-cause mortality [19]. Similarly, Patel et al. [21] observed no association between BMI and recurrence or survival in a cohort of New Zealand patients with early-stage disease. Céspedes et al. [22] also found no significant association between BMI and survival after adjusting for molecular subtype, although luminal A tumors combined with class II/III obesity were linked to poorer outcomes. These heterogeneous findings suggest that BMI alone may be an insufficient surrogate for nutritional status and that more integrative biomarkers may better predict prognosis.

In our study, the significant associations of hemoglobin and albumin with overall survival underscore the importance of metabolic and systemic nutritional indicators. Cellular iron is an essential nutrient; however, cancer cells exhibit dysregulated iron uptake, storage, and export, leading to increased oxidative stress and DNA damage [20,22,23]. Elevated estrogen concentrations can further disrupt iron metabolism, promoting carcinogenesis [24,25]. Additionally, iron may regulate oncogenic signaling pathways such as WNT and JAK–STAT3, contributing to tumor proliferation and metastasis [26,27].

Due to subgroups with small sample sizes because there are missing data in some variables, significant differences in overall survival may not have been found in some variables.

Serum albumin, a key marker of nutritional and inflammatory status, has consistently been linked to cancer outcomes. Reduced albumin concentrations may reflect chronic inflammation, hepatic dysfunction, or systemic malnutrition, all of which can impair treatment tolerance and immune response. Similarly, previous studies have identified combined indices, such as the HALP score (hemoglobin, albumin, lymphocyte, and platelet), as predictors of poor prognosis and chemoresistance in breast cancer, particularly in triple-negative subtypes [28]. Furthermore, lower pretreatment albumin levels and elevated platelet counts have been associated with reduced disease-free survival (DFS), potentially through pathways of systemic inflammation and altered hematopoiesis [27].

A principal limitation of this study is its retrospective design, as it relied on clinical records from a 2017 cohort. Consequently, several potential confounding factors, such as treatment heterogeneity and the absence of longitudinal follow-up data, could not be fully controlled. Nonetheless, the incorporation of official survival outcomes obtained from the national registry (RENIEC) strengthens the validity and accuracy of the endpoint assessment. Future prospective studies with larger, stratified cohorts and comprehensive molecular profiling are warranted to validate these findings and to further elucidate the relationship between nutritional biomarkers and patient survival.

## Conclusions

In this study, anthropometric variables show that most patients are overweight, with very high risk according to waist circumference with no caloric malnutrition and no protein malnutrition. Biochemical variables indicate that most p, have normal albumin levels months were estimated at 93%, 84%, and 79%, respectively. No significant difference was found in overall survival according to anthropometric variables in this group of patients. However, a significant difference was found in overall survival according to biochemical variables: hemoglobin, albumin, and total lymphocyte count. These variables also had a significant effect on time to death.

Heterogeneous findings suggest that BMI (overweight and obesity) alone may be an insufficient surrogate for nutritional status and that more integrative biomarkers or indices may better predict prognosis.

## Acknowledgments

We thank the Instituto Nacional de Enfermedades Neoplásicas from Perú for supporting the execution of the study.

## Author contributions

**Conceptualization:** Lourdes Sánchez-Saldaña, José Cotrina-Concha, Michelle Lozada-Urbano, Jaime Rosales-Rimache.

**Formal analysis:** Yasser Sullcahuaman-Allende, José Cotrina-Concha, Marco Velarde-Méndez, Jorge Chavez-Chocano, Enrique Rodríguez-Coyla, Luis Zambrano-Jaimes, Raúl Mantilla-Quispe.

**Investigation:** Lourdes Sánchez-Saldaña, Yasser Sullcahuaman-Allende, José Cotrina-Concha, Marco Velarde-Méndez, Jorge Chavez-Chocano, Enrique Rodríguez-Coyla, Luis Zambrano-Jaimes, Raúl Mantilla-Quispe, Michelle Lozada-Urbano.

**Methodology:** Lourdes Sánchez-Saldaña, Yasser Sullcahuaman-Allende, José Cotrina-Concha, Marco Velarde-Méndez, Jorge Chavez-Chocano, Enrique Rodríguez-Coyla, Luis Zambrano-Jaimes, Raúl Mantilla-Quispe, Michelle Lozada-Urbano.

**Project administration:** Lourdes Sánchez-Saldaña, Jorge Chavez-Chocano.

**Resources:** Yasser Sullcahuaman-Allende, José Cotrina-Concha, Marco Velarde-Méndez, Enrique Rodríguez-Coyla, Luis Zambrano-Jaimes, Raúl Mantilla-Quispe, Michelle Lozada-Urbano.

**Supervision:** Jaime Rosales-Rimache.

**Validation:** Jaime Rosales-Rimache.

**Visualization:** Jaime Rosales-Rimache.

**Writing – original draft:** Lourdes Sánchez-Saldaña, Yasser Sullcahuaman-Allende, José Cotrina-Concha, Marco Velarde-Méndez, Jorge Chavez-Chocano, Enrique Rodríguez-Coyla, Luis Zambrano-Jaimes, Raúl Mantilla-Quispe, Michelle Lozada-Urbano.

**Writing – review & editing:** Michelle Lozada-Urbano, Jaime Rosales-Rimache.

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
