## [Decision Letter · Decision Letter 0]

20 Jun 2025

Dear Dr. Rosales-Rimache,

Thank you for submitting your manuscript to PLOS ONE. After careful consideration, we feel that it has merit but does not fully meet PLOS ONE’s publication criteria as it currently stands. Therefore, we invite you to submit a revised version of the manuscript that addresses the points raised during the review process.

We look forward to receiving your revised manuscript.

Kind regards,

Shibajee Debbarma, M.D.

Academic Editor

PLOS ONE

Journal Requirements:

“Acknowledgments: We thank the National Institute of Neoplastic Diseases of Peru for providing authorization and the necessary facilities to obtain information from medical records, and Universidad Científica del Sur for the payment of APC.”

3. In the online submission form, you indicated that:

“The data that support the findings of this study are available from the corresponding author upon reasonable request.”

5. Please ensure that you include a title page within your main document. You should list all authors and all affiliations as per our author instructions and clearly indicate the corresponding author.

6. Please amend either the abstract on the online submission form (via Edit Submission) or the abstract in the manuscript so that they are identical.

**Additional Editor Comments:**

After extensive reviews by a wide range of reviewers, it has been concluded that the manuscript demands "Major Revision". Reviewers' comments have been included in this mail. Authors are requested to go through the comments thoroughly and submit the revised manuscript for further consideration for publication.

Reviewers' comments:

Reviewer's Responses to Questions

**Comments to the Author**

1. Is the manuscript technically sound, and do the data support the conclusions?

Reviewer #1: No

Reviewer #2: Partly

Reviewer #3: Partly

Reviewer #4: Yes

Reviewer #5: Yes

Reviewer #6: Yes

Reviewer #7: Yes

Reviewer #8: Partly

2. Has the statistical analysis been performed appropriately and rigorously?

Reviewer #1: No

Reviewer #2: No

Reviewer #3: Yes

Reviewer #4: Yes

Reviewer #5: Yes

Reviewer #6: Yes

Reviewer #7: Yes

Reviewer #8: Yes

3. Have the authors made all data underlying the findings in their manuscript fully available?

Reviewer #1: No

Reviewer #2: Yes

Reviewer #3: Yes

Reviewer #4: Yes

Reviewer #5: No

Reviewer #6: Yes

Reviewer #7: No

Reviewer #8: Yes

4. Is the manuscript presented in an intelligible fashion and written in standard English?

Reviewer #1: No

Reviewer #2: No

Reviewer #3: Yes

Reviewer #4: Yes

Reviewer #5: No

Reviewer #6: Yes

Reviewer #7: Yes

Reviewer #8: Yes

Reviewer #1: General Comments:

This manuscript presents results from a retrospective cohort study aimed at assessing associations between anthropometric measures and biochemical measures of malnutrition with overall survival in breast cancer patients. The authors reported no significant associations between anthropometric measures and overall survival, but did find significant associations between certain biochemical measures of malnutrition such as irone, albumin and total lymphocyte with overall survival. Overall, the topic of this paper is of interest, but important details and justifications are missing throughout (e.g., on the inclusion of important covariates and use of categorical rather than continuous variables). Specific comments for each section and provided below.

Abstract

1. Please define “survival differences”. Is this based on recurrence, all-cause mortality, cancer-specific mortality, etc.

2. Please add more details on the variables that were included in the univariate and multivariate analyses. Furthermore, please add details on the biochemical and nutritional measures that were included in the analysis. Was weight gain also included, given the initial sentence of the abstract?

3. Something is missing in the concluding sentence “specific biochemical markers” Furthermore, did the authors consider looking at diet quality as a predictor given the conclusion about food quality control? It is unclear based on the abstract which predictors were included for diet quality or biochemical markers of nutrition. Similarly, more details on the included anthropometric data would be appreciated.

Introduction

4. The third paragraph in the introduction talks about breast cancer risk and survival outcomes interchangeably. Similarly, the authors present evidence on diet measures with quality of life but not cancer progression or survival outcomes, but on the other hand, talk about the risk of obesity for cancer survival outcomes (but not quality of life). Hence, the authors should revise this paragraph to streamline these ideas, so the reader has a better understanding of the topic of this paper (i.e., prevention, survival outcomes or quality of life outcomes?)

5. More details are needed for the aims. For instance, which measures are included or define “nutritional status”? Which survival outcomes are included? The authors mention anthropometric measures in the abstract, but these are not part of the aims (and please specify which anthropometric measures are included). What are the authors’ hypotheses?

Methods

6. More details are needed to define the outcome. For instance, how was overall survival defined and what criteria were used. Furthermore, is overall survival the same as all-cause mortality?

7. The authors mention consideration for breast cancer subtype in the analyses. How were these groups defined and were there enough participants per group to consider BC subtype? Furthermore, did the authors consider any other variables as potential covariates?

8. The authors mention qualitative variables, they likely meant quantitative variables?

9. The authors need to further explain or define which covariates were included in their Cox proportional hazards models. Further justification for the use of categorical rather than continuous anthropometric and biochemical variables is also warranted. Lastly, further details are needed on the grouping of categories, for instance, what was the minimal sample size required for each group to avoid grouping?

Results

10. While recognizing that the original aim was to look at weight gain/obesity as a risk factor for survival, many of the significant findings were related to indicators of underweight or malnutrition. Therefore, it may be beneficial to revise the paper accordingly and focus on nutritional indicators of malnutrition or underweight as being a risk factor for low survival (rather than overweight and obesity).

11. It is not evident which covariates were controlled for in these associations or how the BC subtypes were used in analyses. Are these results from crude unadjusted models or were covariates considered? Furthermore, adjusting for body weight in the nutritional markers with survival outcomes may provide further insight into these associations.

Discussion

12. Overall, the discussion reads a bit disjointed. There are multiple paragraphs and few transitions between ideas. While the authors do mention findings from several studies, there is little interpretation of these findings or comparison to their own findings.

13. A conclusion on further directions for this work is needed.

Reviewer #2: - The methodological part needs to be revised; the description of the instruments used for anthropometric (type of scale, plicometer, ect) and biochemical measurements is missing.

- The statistical analysis is unclear and needs to be revised. The mean was used but without the standard deviation, and some data are presented as medians. The sample distribution and the choice of parametric and/or nonparametric analysis would need to be clarified in this regard. Also, I do not understand the meaning of min and max and even less clear the use of 1 quartile and 3 quartile (usually 1 vs. 4 is used). Justify the choices! Finally, no statistical analysis of comparison was used in the presentation of these data, which makes the quartile division less clear.

- The intervals of the variables should refer to reference cutoffs but no literature reference was reported

- In the results I think it is better to present the data in mean ± standard deviation rather than the range. Also, it would have been interesting to see how these anthropometric and biochemical variables changed from 12m to 60m.

- Iron is mentioned a lot in the discussion but this should be better described in the introduction and its link to anemia as the latter is not always related to iron deficiency.

- The most obvious results of the study from the Cox regression analysis were not well argued in the discussion without proper comparison with other studies.

Reviewer #3: The research article examines the association between anthropometric and biochemical nutritional indicators and survival outcomes in women with breast cancer in Peru. However, my concerns/suggestions are not necessarily a reflection of the quality of the work as the following:

• Although the manuscript mentions 215, or possibly 195, breast cancer (BC) cases were reportedly observed, as mentioned in the introduction, the aim to determine the association between nutritional status and survival in women with breast cancer seems to be a superficial rationale for the study. The current research lacks depth and fails to present novel insights into the function, risk assessment, or biology of these cancer trials. Additionally, the number of reported cases is inconsistently presented and unclear in the text; please double-check and ensure consistency throughout

• Although this is a minor issue, the authors should ensure that all abbreviations are clearly defined upon their first appearance in the text. Additionally, there are formatting inconsistencies, particularly with the repeated use of the acronym "BC" for "breast cancer" without reintroducing the full term and vice versa. For instance, "breast cancer" is mentioned at the beginning of both the Introduction and Methodology sections, but only the abbreviation is used subsequently. This may confuse readers unfamiliar with the acronym and disrupt the flow of the text. So, please double-check.

• Another minor issue is the inconsistent formatting of citations in the Word version of the manuscript. Some references were entered manually, while others were generated using citation software. I recommend enabling automatic citation formatting and reviewing all references to ensure consistency and adherence to the required style guidelines.

• Some comments about figures and tables here and there!

o In Figure 1:

Lack of sample size (n=?) indication anywhere in the figure or legend, making percentage interpretation incomplete.

There’s no statistical analysis mentioned, so interpretation is purely descriptive.

o In Table 1:

This table presents the essential demographic and clinical characteristics of 195 patients, providing insights into the population structure, disease subtype, and treatment modalities.

• Missing Data (“No registration”) as several critical clinical variables have high rates of missing data, such as histological type: 26.7% missing.

• Inconsistent Terminology such as “Partner” vs. “Married” may confuse readers. Are “Partner” and “Married” mutually exclusive? If not, merge or redefine.

o In Figure 2:

No statistical context (e.g., means, p-values) or demographic breakdown is offered in Figure 1.

There are no defined clinical cutoffs and classification systems.

o In Table 2:

Inconsistent units.

The abbreviation "Size" is not standardized, as clinical literature rarely uses "Size" alone.

o In Table 3:

Again, inconsistent units.

o In Table 4:

Missing statistical test mention.

Small sample size in subgroups.

Lack of clarity in group definitions.

• From a technical perspective, the authors state in the Discussion that their analysis included 195 women and found no significant association between anthropometric measures and overall survival. They further rely on existing literature, which includes data from several thousand cases, to support these findings. However, this comparison highlights the limitations of their results, given the relatively small sample size in the current study. Moreover, the authors have not provided a thorough, critical analysis or a deeper Discussion of how their findings relate to or differ from the broader literature.

• The Results, Discussion, and Conclusion sections require improvement by incorporating more precise molecular and cancer biology terminology. Currently, the authors appear to focus primarily on describing their technical and statistical methods, as well as summarizing literature, without providing a deep biological analysis or critical interpretation of their findings. Strengthening the biological rationale and offering a more thorough critique would enhance the overall impact and scientific rigor of the study.

Reviewer #4: General Comments

The study explores a correlation that has been extensively debated in the literature, yet definitive conclusions remain elusive. Research of this kind is indeed necessary, particularly regarding the impact of nutritional status on malignant pathologies, including breast cancer. Personalized management strategies should incorporate nutritional assessment and interventions. While the study addresses a relevant and timely topic, the methodology lacks sufficient rigor to support clinically meaningful conclusions. A longitudinal assessment of nutritional status would have been valuable in evaluating its evolution alongside the progression of malignant disease.

Introduction

The introduction is well-written and clearly structured, providing an appropriate context for the study.

Methodology and Results

• It is unclear why the inclusion period for the study was limited to only one month. Please provide justification for this short timeframe.

• Were the anthropometric data extracted retrospectively from existing medical records, or were patients invited for a dedicated clinical assessment specifically for the study?

• At what point in the clinical course were the anthropometric parameters measured (e.g., at diagnosis, during treatment, or post-treatment)?

• The methodology section states that 215 records were included, while the results section reports data on only 195 participants. This discrepancy should be clarified.

Discussion and References

The discussion contextualizes the findings within the existing literature. However, many of the cited references are outdated. More recent and relevant literature should be included to strengthen the discussion and align it with the current state of research.

Conclusions

The conclusions are appropriately drawn based on the presented results.

Reviewer #5: • Double check author guidelines and format your manuscript to meet these guidelines.

• There are several grammatical errors in your manuscript. Kindly check and address them.

• Your introduction needs restructuring. There is not enough information on available evidence regarding your research question, and the gap(s) identified which may justify the conduct of this study.

• You include a table on molecular subtypes and proliferation index, yet this table was not referenced in the preceding paragraph.

• “A descriptive analysis of qualitative variables…” You mean categorical variables?

• What guided the grouping of the anthropometric and biochemical variables?

• Discussion is fine, but I think the first 4 paragraphs focus too much on reporting results of other studies and not really synthesizing your findings. Why do you think non-significant associations were found in your study?

• Your paragraph on the study’s limitations is not clear.

• Your conclusion is just a summary of your results. What is the take-home message for readers?

• What recommendations and/or directions do you have for future research?

Reviewer #6: The authors made good efforts in producing this important study. The only challenge is that it is difficult to associate the survival of BC patients with variables such as anthropometric and biochemical variables in that there are other issues for instance, economic problems, death due to unable to come to health facility due to various reasons. Moreover, it is difficult to conclude that those patients who were in loss of follow up are died. Thus, I wonder if the authors consider these issues in their study and put some clarifications.

Reviewer #7: PLOS ONE Peer Review Decision Letter

Manuscript ID:PONE-D-24-55103

Title: Anthropometric and Biochemical Nutritional Indicators and Survival in Women with Breast Cancer: A Retrospective Cohort Study

Journal: PLOS ONE

Decision: Major Revisions Required

Dear Authors,

Following a thorough examination based on PLOSONE journal criteria, recent literature, reference consistency, and potential concerns about originality and scientific rigor, I have discovered many strengths and shortcomings that must be addressed before further consideration.

1. Scientific and Methodological Assessment

- Scope Fit: The manuscript addresses important nutritional and biochemical predictors of survival in breast cancer—which aligns well with PLOS ONE’s focus on scientifically sound studies regardless of novelty. The study design demands interdisciplinary

- Design Strengths: Retrospective cohort design, appropriate statistical tools (Kaplan-Meier, Cox regression), and a clearly defined Peruvian patient population.

- Limitations: Anthropometric data were not prospectively measured nor linked to dynamic nutritional interventions. Important covariates (e.g., comorbidities, socioeconomic status, treatment protocols) are missing. Reference ranges for albumin (e.g., reported up to 462 g/L, which is not physiologically plausible) suggest potential data entry or unit errors.

2. Comparison of Recent Literature

Several studies have investigated comparable subjects. The manuscript mentions many of this research, however there is no critical comparison commentary in the commentary section. Consider racial/genetic differences and healthcare access while discussing your findings.

3. Reference Audit

Most references are relevant and up to date. However, there are some small errors that should be addressed, such as a missing DOI/PMID and proper formatting in accordance with the PLOS ONE citation standard.

4. Plagiarism and Authorial Integrity

There's no indication of plagiarism. However, closely paraphrased portions require a more distinct rephrasing with appropriate attribution.

5. Ethics & Transparency

Ethics approval is documented. However, data access is still inadequate under PLOS ONE's open data policy. Authors must upload anonymized data as supplemental data.6. Additional improvements are required.

- Check and Clarify the albumin levels; 462 g/L is not biologically possible. Please have a check on other biochemical values too.

- Improve the abstract and conclusion for clarity. Address sentence fragmentation and missing transitions.

- Include full figure legends underneath each figure.

- Explain any missing data and how it may affect the conclusions (for example, subtype data or histology type).

Summary Table

Category Status

Scientific Soundness Acceptable but limited by retrospective design

Novelty Slightly Limited or under expressed—requires clearer distinction from prior studies

Reference Accuracy Requires minor revisions

Statistical Rigor Acceptable

Plagiarism Check Passed

PLOS ONE Policy Compliance Data availability statement must be revised

Final Verdict Major and minor Revisions Required

Final Recommendation : Substantial revisions to improve scientific lucidity, guarantee complete data transparency, and fortify comparative analysis with existing literature. Provided that the editorial board and peer reviewers concur with this assessment, this manuscript has the potential for publication in PLOS ONE upon satisfactory revision, with a particular emphasis on data availability, reference consistency, and core analytical issues.

Reviewer #8:

1. Is the manuscript technically sound, and do the data support the conclusions?

i. In the manuscript rationale of the study is not written properly. It would be better to write it properly mentioning why authors aimed to conduct this study.

ii. All the findings of the study have not been discussed and justified in the discussion section of the manuscript. It would be better to discuss the findings of the study and justify them.

iii. Yes, the data support the conclusions of the study. It would be better to remove percentage values given in the conclusion.

iv. Conclusion of the abstract is not written properly. It would be better to write based on the findings of the study

v. It would be better to modify the title of the study as "Association of Anthropometric, Biochemical, and Nutritional Indicators with Survival in Women with Breast Cancer: A Retrospective Cohort Study"

vi. At several places in the manuscript nutritional status is written. In place of nutritional status, it would be better to write nutritional indicators, since nutritional status has not been assessed in the study.

*2. Has the statistical analysis been performed appropriately and rigorously?

Yes, however, units of some of the variables are missing in the text written under "statistical analysis".

*3. Have the authors made all data underlying the findings in their manuscript fully available?

Yes

*4. Is the manuscript presented in an intelligible fashion and written in standard English?

Yes, however, there are grammatical errors in the abstract. Please check and correct them.

**Do you want your identity to be public for this peer review?** For information about this choice, including consent withdrawal, please see our Privacy Policy

Reviewer #1: No

Reviewer #2: No

Reviewer #3: No

Reviewer #4: **Yes:** Andronic Octavian

Reviewer #5: No

Reviewer #6: **Yes:** Tamiru Demeke

Reviewer #7: No

Reviewer #8: No

---

## [Author Response · Author response to Decision Letter 1]

11 Dec 2025

We thank the Editor and reviewers for their thorough evaluation and constructive comments. We have carefully revised the manuscript in accordance with all suggestions. A detailed, point-by-point response to each reviewer and editor comment is provided in the attached rebuttal letter, indicating the changes made and where they can be found in the revised manuscript.

We remain at your disposal for any further clarification or additional modifications that may be required.

---

## [Editor Report · Decision Letter 1]

8 Jan 2026

Anthropometric and Biochemical Nutritional Indicators and Survival in Women with Breast Cancer: A Retrospective Cohort Study

PONE-D-24-55103R1

Dear Dr. Rosales-Rimache,

We’re pleased to inform you that your manuscript has been judged scientifically suitable for publication and will be formally accepted for publication once it meets all outstanding technical requirements.

Kind regards,

Shibajee Debbarma, M.D.

Academic Editor

PLOS One
---

## [Editor Report · Acceptance letter]

PONE-D-24-55103R1

PLOS One

Dear Dr. Rosales-Rimache,

I'm pleased to inform you that your manuscript has been deemed suitable for publication in PLOS One. Congratulations! Your manuscript is now being handed over to our production team.

Kind regards,

on behalf of

Dr. Shibajee Debbarma

Academic Editor

PLOS One